# Global Impact of COVID-19 on Colorectal Cancer Screening: Current Insights and Future Directions

**DOI:** 10.3390/medicina58010100

**Published:** 2022-01-10

**Authors:** Jonathan Kopel, Bojana Ristic, Gregory L. Brower, Hemant Goyal

**Affiliations:** 1Department of Medicine, Texas Tech University Health Sciences Center, Lubbock, TX 79430, USA; 2Cell Biology and Biochemistry, Texas Tech University Health Sciences Center, Lubbock, TX 79430, USA; bojana.ristic@outlook.com; 3Department of Medication Education, Texas Tech University Health Sciences Center, Lubbock, TX 79430, USA; greg.brower@ttuhsc.edu; 4The Wright Center for Graduate Medical Education, Scranton, PA 18501, USA; goyalh@thewrightcenter.org

**Keywords:** colorectal cancer screening, COVID-19, SARS-CoV-2

## Abstract

The coronavirus disease 2019 (COVID-19) pandemic has brought significant challenges to many aspects of healthcare delivery since the first reported case in early December 2019. Once in the body, SARS-CoV-2 can spread to other digestive organs, such as the liver, because of the presence of ACE2 receptors. Colorectal cancer (CRC) remains the second-leading cause of death in the United States (US). Therefore, individuals are routinely screened using either endoscopic methods (i.e., flexible sigmoidoscopy and colonoscopy) or stool-based tests, as per the published guidelines. At the beginning of the COVID-19 pandemic, the Centers for Medicare and Medicaid Services (CMS) recommended that all non-urgent surgical and medical procedures, including screening colonoscopies, be delayed until the pandemic stabilization. This article aims to review the impact of COVID-19 on CRC screening.

## 1. Introduction

The coronavirus disease 2019 (COVID-19) pandemic has brought significant challenges to many aspects of healthcare delivery since the first reported case in early December 2019 [1]. SARS-CoV-2 predominantly affects lungs, causing pneumonia, and acute respiratory distress syndrome (ARDS), but can also have extrapulmonary involvement, particularly with gastrointestinal (GI) symptoms [2]. GI symptoms, such as diarrhea (2–10.1%), nausea, and vomiting (1–3.6%), occur with modest frequency in COVID-19 compared to the fever and pulmonary symptoms in most patients [3,4,5]. Although the pathogenesis is still being investigated, current data suggests that the primary step for SARS-CoV-2 entry into the enterocytes occurs via the angiotensin-converting enzyme 2 (ACE2) protein [6,7,8,9,10,11].

Colorectal cancer (CRC) remains the second-leading cause of death in the United States (US) [12]. Therefore, individuals are routinely screened using either endoscopic methods (i.e., flexible sigmoidoscopy and colonoscopy) or stool-based tests, as per the published guidelines [13,14,15,16,17,18]. In the United States, the U.S. Preventive Services Task Force (USPSTF) recommends screening for colorectal cancer in all adults aged 50 to 75 years as, well as for adults aged 45 to 49 years [18]. In addition, the USPSTF recommends that clinicians selectively offer screening for colorectal cancer in adults aged 76 to 85 years. At the beginning of the COVID-19 pandemic, the Centers for Medicare and Medicaid Services (CMS) recommended that all non-urgent surgical and medical procedures, including screening colonoscopies, be delayed until the pandemic stabilization [19]. This was also done to reduce the potential risk of exposure to SARS-CoV-2, given the virus is present in fecal matter from COVID-19 patients [1,20,21]. In response, there was a 90% decrease in CRC screenings, resulting in a 32% decrease in new CRC diagnoses, as well as a 53% decline in CRC-related surgical procedures by mid-April 2020 [19]. Moreover, by April 2021, the routine screening colonoscopy rate remained 50% lower than the pre-pandemic times [19]. Other forms of cancer screening, such as mammograms and pap tests, also decreased during the pandemic [22,23]. This article aims to review the impact of COVID-19 on CRC screening.

## 2. Methods

A literature search was performed using PubMed, Embase, SCOPUS, OVID, and Web of Science databases up to June 2021 to identify articles related to CRC screening and COVID-19. The search words used were “colorectal cancer”, “screening”, “COVID-19”, and “SARS-CoV-2” alone and in combination. The results of the search are shown in Figure 1. The inclusion criteria for the studies included retrospective, longitudinal, and randomized control studies using different colorectal screening methods (guaiac-based fecal occult blood test, fecal immunochemical test (FIT), FIT-DNA test, colonoscopy, and sigmoidoscopy) during the COVID-19 pandemic. The exclusion criteria included any studies that were case reports, supplements, abstracts, commentaries, or had the wrong study focus for this review. A total of 331 studies were identified after the initial search. On initial review of the title and abstracts, 242 manuscripts were excluded because of irrelevance to our topic of interest. A further full manuscript review of the remaining articles was performed. A total of 20 articles were finally included in this review after removing 4 abstracts, 1 supplement, 23 commentaries, and 5 irrelevant studies.

## 3. Effects of COVID-19 Pandemic on CRC Screening Pathway

The start of the COVID-19 pandemic disrupted several levels of primary care in the prevention of several preventable diseases, including CRC. Given the fear of transmitting SARS-CoV-2 in hospital settings, many elective procedures, such as colonoscopies, were discontinued until further notice. As a result, most primary care physicians utilized fecal immunochemical testing to continue providing CRC screening to prevent patients from traveling to the hospital and potentially exposing themselves to SARS-CoV-2. Amidst the COVID-19 pandemic, the CRC screening program, which included initial fecal immunochemical testing (FIT) and diagnostic endoscopy, remained fully functional in the National Taiwan University Hospital in Northern Taiwan [24]. However, in comparison to the screening data collected from the corresponding quartal for previous years, the operations in this screening hub were interrupted [24]. This was observed due to the significant reduction in the number of patients that participated in the FIT screening, followed by the decrease in the immediate referrals to the diagnostic colonoscopy for the FIT-positive patients [24]. In addition, the already scheduled colonoscopy appointments had higher cancellation and rescheduling rates, with patients often listing the fear of nosocomial COVID-19 infection as the reason for not undergoing the procedure [24]. Furthermore, the number of diagnostic colonoscopies decreased drastically at the screening center in Japan during the state of emergency, which lasted 120 days [25]. The number of performed CRC surgeries, however, remained unchanged from the ones during previous years [25]. In comparison to the corresponding time period for three years prior to the COVID-19 pandemic, the population-based study in Hong Kong reported a 58.8% reduction in the number of lower colonoscopies performed from October 2019 to March 2020, which resulted in a 37% decline in the diagnosis of novel CRC cases [26].

The low to middle-income countries that recently started implementing nationwide CRC screening programs, such as Paraguay, Thailand, Iran, and Malaysia, continued to offer CRC screening, diagnostic, and treatment procedures during the COVID-19 pandemic [27]. They reported that administrating screening tests and diagnostic services for the screen-positive individuals were from 20–90% of the pre-COVID times, whereas the status of treatment services for cancer patients was better, with 65–90% ratings [27]. The authors discussed that the ability of these screening programs to resume to full capacity or beyond would be significantly difficult for the low to middle-income countries than it would be for the high-income countries [27]. Thus, the pandemic will have a lasting effect on these programs [27].

Most of the facilities that offered CRC services in England and Wales recorded a reduction in patient referral, resulting in an alteration in the CRC treatment plans that were brought up either by the delayed start of the treatment due to the fear of infection, lack of tissue diagnosis and radiological staging, and/or the limited resources [28]. The population-based study conducted in England revealed that the COVID-19 lockdown for months of April, May, and June 2020 reduced the CRC diagnostic rates, as the 2-week-wait referrals significantly dropped by 23%, whereas the number of performed colonoscopies decreased by 46% [29]. Furthermore, the lockdown elicited a 19% reduction in implementing the appropriate 31-day treatment plans [29]. Due to this, this study reported that over 3500 patients missed early CRC diagnosis, and did not undergo potentially lifesaving procedures during the COVID-19 lockdown [29]. The CRC screening rates remained low during the lockdown even, when that was performed in the “COVID-19 free” facility, which did not admit COVID-19 positive patients, and ensured ongoing COVID-19 testing for the facility’s staff and patients [30]. The COVID-19 pandemic also affected molecular diagnostic metastatic CRC testing, which included performing quantitative PCR and next-generation sequencing (NGS) for *KRAS*/*NRAS* hot spot mutations in the largest molecular diagnostic centers for cancer patients and high-risk individuals in Serbia [31]. The number of performed analyses for the metastatic CRC during the state of emergency drastically decreased by 46%, followed by a 15% reduction in the number of GI tract cancer patients presented to the tumor board for formulating and implementing further treatment plans [31]. This trend did not recover even after the state of emergency was lifted [31].

The total deficit for CRC screening in the US during the COVID-19 pandemic compared to 2019 was calculated to be 3.8 million cases for both men and women [32]. The screening rates, when compared to 2019, decreased drastically by 79.3% in April of 2020 as a result of the lockdown, and they started to increase in June and July; the increase, however, did not re-calibrate to the same number as it was in 2019/2018, which showed that compensation for the missed diagnosis was not possible at the moment (13.1% lower than in previous years) [32]. The sharpest decline in CRC screening during lockdown was recorded in the Northeast geographical region of the USA, and amongst the population with a higher socioeconomic status [32]. In part, this was due to a 43% reduction in total referrals of primary care, with a 79% decline in urgent referrals, 64% reduction in routine referrals, and 40% reduction in “urgent suspicion of cancer” (USOC) patient referrals [33]. Moreover, during lockdown months in a healthcare system in Los Angeles, the colonoscopy rates declined to about only 12 per week, compared to the 223 per week pre-pandemic, whereas the FIT dropped to 61 per week, compared to 154 per week pre-pandemic [33]. After the lockdown was lifted, the number of colonoscopy appointments recovered to the number before the pandemic, whereas the noninvasive FIT and stool DNA tests recovered, and exceeded the pre-pandemic numbers [34]. Lastly, the interruption in developing screening programs in the CRC “hotspot” Appalachian Kentucky region elicited a significant backlog, and furthered the barriers that the program already had to face during its development [35].

## 4. The Aftermath of the Halted CRC Screening Due to COVID-19

The immediate halt in CRC screening and diagnostic modalities as a response to the COVID-19 pandemic has already been reflected in the sudden decrease in the diagnosis of the novel CRC cases [26,29,32]. Attributable to the interruption in timely diagnostic colonoscopy, the number of urgent admissions ascribed to obstructive CRC significantly increased during this time [25]. Moreover, the detected CRC cases were more severe in the lockdown group than in the previous years (47% vs. 25%), as the high-risk adenomas were often larger than 10 mm, contained villous compartment, high-grade dysplasia, and were serrated. However, the low-risk adenoma detection rate decreased (9% vs. 22%). Altogether, the lockdown group exhibited higher CRC detection (8% vs. 1%) [30].

The interruption of the diagnostic colonoscopies will also elicit long-term effects. Lui et al. predicted that 6.4% of CRC would have higher stage shifting, with an increase in stage IV carcinomas [26]. If the reduction in diagnostic procedures was not met, and if it further reduced to 20%, the stage shifting would increase by 7.2% [26]. The four country-specific CRC microsimulation models revealed that the delay in CRC diagnosis due to the diagnostic interruptions of three, six, and twelve months would result in a significant increase in the number of CRC incidence and CRC-related deaths during the period between 2020 and 2050 [36]. It was estimated that the relative increase of twelve-month disruptions would result in 0.4–0.9% additional CRC cases and 0.8–1.2% additional CRC-related deaths in the Netherlands, 1.2% additional CRC cases and 2% additional CRC-related deaths in Australia, and 0.6% additional CRC cases and 0.8% additional CRC-related deaths in Canada [36]. This devastating statistic can be minimized and mitigated if urgent catch-up screenings are provided to the screening and diagnostic facilities [36]. The delays also existed in performing the diagnostic colonoscopy for the two-week referral patients who were flagged urgent due to positive FIT results and severe symptoms [37]. The UK modeling study estimated that failure to provide timely diagnostics to these patients attributable to the delays of two, four, and six months in the follow-up diagnostic colonoscopies will significantly increase CRC-related deaths, and will result in a loss of life years [37].

A recent study by Santoro et al. studied the global impact of the COVID-19 pandemic on CRC screening, using a 35-item survey to assess the impact of COVID-19 on preoperative assessment, elective surgery, and postoperative management of colorectal cancer patients [38]. Respondents were sorted into two groups for comparison: (1) “delay” group: pandemic-affected colorectal cancer care; and (2) “no delay” group: unaffected colorectal cancer treatment. A total of 1051 respondents from 84 countries completed the survey [38]. There were no significant variations in demographics between the delay (*n* = 745, 70.9%) and no delay (*n* = 306, 29.1%) groups. In the delayed group, 48.9% of respondents reported a change in the initial surgical plan, and 26.3% reported a shift from elective to urgent operations [38]. Reductions in interdisciplinary team meetings, and the relocation of hospital and staff resources were significantly associated with delays in endoscopy, radiology, surgery, histopathology, and prolonged chemoradiation therapy-to-surgery intervals [38]. Furthermore, the status of the epidemic was linked to a patient’s overall recovery during colorectal cancer treatment. Overall, there were noticeable improvements in colorectal cancer diagnostic and treatment procedures across the world. Rather than geographic variables, changes in CRC screening were linked to disparities in health care delivery systems, hospital preparation, resource availability, and local coronavirus illness 2019 prevalence [38].

A similar study conducted in the United Kingdom by the COVIDSurg Collaborative investigated the impact of SARS-CoV-2 on mortality after surgical resection of CRC during the early stages of the COVID-19 pandemic on surgical practice [39]. The COVIDSurg Collaborative used an international cohort study of patients who had colon or rectal cancer, and were not suspected of having SARS-CoV-2 before surgery. Using 2073 patients from 40 nations, the study found that 1.3 percent (27/2073) had a defunct stoma, and 3.0 percent (63/2073) had an end stoma rather than just an anastomosis. Thirty-day mortality was 1.8 percent (38/2073), with a 3.8 percent (78/2073) incidence of postoperative SARS-CoV-2, and a 4.9 percent (86/1738) anastomotic leak rate [39]. Patients without a leak or SARS-CoV-2 had the lowest mortality rate (14/1601, 0.9%), whereas patients with both a leak and SARS-CoV-2 had the greatest mortality rate (5/13, 38.5%) [39]. In contrast, anastomotic leak (adjusted odds ratio 6.01, 95 percent confidence interval 2.58–14.06), postoperative SARSCoV-2 (16.90, 7.86–36.38), male sex (2.46, 1.01–5.93), age > 70 years (2.87, 1.32–6.20), and advanced cancer stage (3.43, 1.16–10.21) were all independently linked with mortality [39]. There were fewer anastomotic leaks (4.9 percent versus 7.7%), and an average shorter duration of stay (6 versus 7 days) compared to pre-pandemic data, but increased mortality (1.7 percent versus 1.1 percent) [39]. Based on the patient, operational, and organizational risks, the COVIDSurg Collaborative suggested surgeons should take additional precautions against SARS-CoV-2 and anastomotic leak when performing surgery during the present and future COVID-19 waves.

## 5. Modified CRC Screening Approaches during COVID-19 Pandemic

In order to mitigate the survival decline attributed to the interruption of the CRC screening pathway elicited by the COVID-19 pandemic, public health screening programs need to be restructured [40]. Loveday et al. noted that prioritizing the high-risk CRC patients via FIT triage would alleviate 89% of deaths that would occur due to the lockdown backlog [37]. In turn, this strategy would reduce the nosocomial COVID-19 deaths [37]. Moreover, incorporating FIT screening as a triage tool for the 2-week wait referral patients elicited successful reallocation of the limited resources for high-risk CRC patients [41]. This mitigation approach was predicted to provide the additional CRC screening to approximately 588,800 novel patients, and establish about 2899 new CRC diagnoses, out of which 68.9% would be early-stage [19]. Furthermore, the utilization of other noninvasive stool-based DNA tests was marked effective in identifying high-risk patients, and prioritizing them for diagnostic colonoscopy and CRC treatment [34,35]. Miller et al. designed and implemented a novel CRC triage procedure named “COVID-adapted pathway”, which successfully ameliorated the adverse effects of the diagnostic colonoscopy backlog [33]. According to this, the USOC patients referred by a general practitioner were triaged based on the severity of their symptoms [33]. In particular, patients with high-risk symptoms were triaged to CT with oral contrast and the quantitative FIT (qFIT), patients with low-risk symptoms were triaged with qFIT alone, whereas patients with palpable mass were outpatients [33]. Using this method, the number of detected cancers was similar to the previous year, while keeping the patients negative for COVID-19 [33]. The authors advised that the qFIT testing needed to be repeated twice, two weeks apart, because they noted occasional variations in results [33].

Furthermore, to ensure that the CRC screening continued, these programs needed to adopt new strategies, such as distributing stool-based CRC screening tests via mail, and shifting from paper to digital educational tools [27,35]. Furthermore, this included telehealth, such as teleconsultation of screen-positive individuals, conducting meetings over the phone, and dedicating call/text centers that would facilitate appointment scheduling [27,32,35]. Human factors, such as proper leadership and the allocation of resources, improved the overall outcome [28].

Lastly, the existence of “cold sites”, such as CRC screening hubs, significantly improved the CRC screening and diagnosis pathway [28,30]. “Cold sites” were defined as the COVID-19 free clinics that were ensured via continuous SARS-CoV-2 testing, segregation of the emergency versus elective procedures, and the geographical separation of the COVID-19 facilities [28,30]. The continued screening in COVID-19 free clinics amidst the infectious agent pandemic was not only effective and necessary, but was also marked as safe, as there were no nosocomial infections related to COVID-19 in patients or in medical staff, as the safety procedures were followed closely [30].

## 6. Effects of COVID-19 Pandemic on Other Cancer Screenings

Since the referrals via the 2-week-wait urgent pathway dropped by 84% in the United Kingdom during the COVID-19 state of emergency, the delays of 3 months during the lockdown and the backlog of the referrals would decrease the 10-year survival by 10%, whereas the 6-month delays would decrease it to 30% for patients that suffer from carcinoma of the colon, rectum, esophagus, lung, liver, bladder, pancreas, stomach, larynx, and oropharynx [42]. Additional delays would further reduce the survival capacity while addressing the backlogs, and prioritizing the high-risk patients would mitigate these statistics [42]. Moreover, fewer breast cancer diagnoses were recorded in the Netherlands due to the suspension of the national screening programs [43]. In addition, low to middle-income countries, on average, reported ratings of less or equal to 50% of the pre-COVID-19 capacities by 61.1% of the participants for screening services, 44.4% of participants for diagnostic services, and 22.2% participants for treatment services for breast and cervical cancers [27]. To ensure the continuity of these national screening and diagnostic cancer programs, this study highlighted the necessity of incorporating the public community outreach through the expansion of the telehealth program [27]. Moreover, the COVID-19 pandemic interrupted the molecular diagnostic testing for non-small cell lung carcinoma (*EGRF* mutations) and metastatic melanoma (*BRAF* mutations), and hereditary breast/ovarian cancer (*BRCA1/2* mutations) in the largest molecular diagnostic center for cancer patients in Serbia drastically decreased [31]. The decline did not recover after the state of emergency was lifted [31]. Lastly, lockdown months exhibited a sharp reduction in screening rates for breast (90.8%) and prostate cancer (63.4%) [32]. It was estimated that about 3.9 million women were undiagnosed for breast cancer due to this interruption, and 1.6 million men missed diagnosis for prostate cancer in the US [32]. The highest decline was recorded for the northwest US region, and amongst the population with a high socioeconomic status [32].

## 7. Clinical Implications and Limitations

There is no doubting that COVID-19 has a significant impact on cancer screening. The CRC death rates are expected to rise as a result of screening test disruptions and delays. Although the outlook for CRC appears bleak, there are several lessons to be learned from the COVID-19 pandemic in terms of CRC screening, diagnosis, and overall prevention. More use of alternate techniques, such as FIT testing, are important for the future of CRC screening and diagnosis. Furthermore, the availability of FIT testing can help to reduce racial health inequalities. Despite the fact that routine screening techniques are beginning to resume as COVID-19 vaccinations are being provided in many industrialized countries, the pandemic has changed the way healthcare practitioners perceive CRC screening. Though colonoscopy will always be the gold standard for CRC screening, FIT tests and other screening procedures offer considerable strengths and unique qualities that make them useful and ideal in certain scenarios. It is critical that we use what we know about COVID-19’s impact on CRC to plan for and prevent human suffering in future pandemics and other public health emergencies. Early detection, combined with adequate care, saves lives, and doing so might help avoid the impact of a future occurrence on CRC from being as disastrous. Regular screening techniques have resumed in recent years, since COVID-19 vaccinations have become more widely available. However, the previously described delays continue to pay a toll on human lives in terms of the delayed diagnosis and progression of CRC.

This study has several limitations. First, the long-term effects of COVID-19 are still being investigated for CRC surveillance and other cancers. The full impact of the pandemic on cancer screening will be understood in the future. Second, the studies did not adequately sample the effects of the COVID-19 pandemic on CRC screening across the globe. This is likely due to differences in severity and resource limitations in acquiring and maintaining data on CRC screening during the pandemic. Third, not all countries equally report CRC screening rates that can be compared between countries. Fourth, different studies reported different methods of CRC screening, which may influence the overall rate of CRC before, during, and after the pandemic. Further longitudinal studies will be needed to address the limitations mentioned in this paper.

## 8. Conclusions

Continuous CRC screening efforts, from population-wide stool-based testing to diagnostic endoscopies and treatments, have elicited early cancer detection, and improved the devastating statistics regarding the CRC diagnosis outcome. However, programs are not available everywhere, and, in some places, are not efficient. Therefore, they require constant improvements in terms of encouraging patient participation, educating the population, and stratifying low- and high-risk patients to ensure cancer diagnosis and prompt treatment. The COVID-19 pandemic caused the world to pause, and instituted lockdowns, notably interrupting CRC screening programs. The reasons for the halt were the allocation of limited hospital resources towards the fight against COVID-19, the ongoing fear of nosocomial SARS-CoV-2 infection, and the overall overwhelming burden that the pandemic placed onto the healthcare system. For CRC screening programs, this included a drop in referrals from a general practitioner, patients’ unwillingness to partake in stool-based testing, canceling or rescheduling colonoscopy appointments by patients out of fear or by institutions because they worked in limited capacities, and changing treatment plans to comply with the pandemic-elicited regulations. Although some centers remained fully functional or adopted novel screening pathway procedures which included telehealth, the diagnostic capacities halted. In this manner, the substantial number of CRC patients went undiagnosed, which, in the short term, resulted in an increase of obstructive CRC, and the presence of high-risk adenomas. The long-term effects of the diagnosis backlog could result in a devastating rise of late-stage CRC cases, and the overall loss of life years due to the lack of appropriate treatments for these patients. These prognostics, however, can be mitigated if proper catch-up screenings are provided. These lessons can also serve as a teaching moment for healthcare leadership, and can provide guidelines for minimizing and altogether avoiding the interruption of cancer screening programs if novel pandemic-causing infectious agents appear.

## Figures and Tables

**Figure 1 medicina-58-00100-f001:**
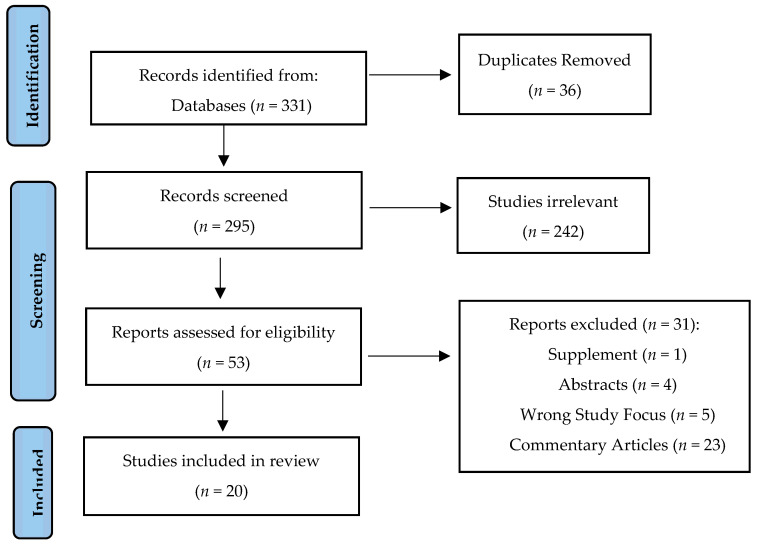
Systematic review article outline.

## Data Availability

Not applicable.

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
