# Peer review of "Global Impact of COVID-19 on Colorectal Cancer Screening: Current Insights and Future Directions"

_medicina, 2022, doi:10.3390/medicina58010100_

Round 1
Reviewer 1 Report
I read your article entitled "Global Impact of COVID-19 on Colorectal Cancer Screening: Current Insights & Future Directions".
I would like to mention a few points to improve the manuscript.
- Provide criteria for selecting and deleting studies in the method section
- Mentioning keywords to select studies may be appropriate
- Is there anything about the infection risks of colonoscopy during the pandemic? (https://www.ncbi.nlm.nih.gov/pmc/articles/PMC7661577/)
Author Response
Reviewer 1:
I read your article entitled "Global Impact of COVID-19 on Colorectal Cancer Screening: Current Insights & Future Directions".
I would like to mention a few points to improve the manuscript.
- Provide criteria for selecting and deleting studies in the method section
We appreciate the reviewer’s comments. We added the appropriate inclusion and exclusion criteria information to the methods section
2. Mentioning keywords to select studies may be appropriate
We appreciate the reviewer’s comments. We added the appropriate inclusion and exclusion criteria information to the methods section. The keywords have been mentioned in the method section of the manuscript
3. Is there anything about the infection risks of colonoscopy during the pandemic? (https://www.ncbi.nlm.nih.gov/pmc/articles/PMC7661577/)
Infectious protocols have been suggested by various international gastroenterology and endoscopy organizations, especially because of the generation of the droplet during endoscopies. Because of the high-risk covid transmission during the procedures, colonoscopies were discouraged during the initial stages of the pandemic. We have added a sentence to the introduction and included the reference mentioned above in the paper.
Reviewer 2 Report
This is a review regarding the impact of COVID-19 on Colorectal Cancer Screening.
The topic has already been addressed but it is still interesting:
Colorectal Cancer Screening: Impact of COVID-19 Pandemic and Possible Consequences. Life (Basel). 2021 Nov 26;11(12):1297. doi: 10.3390/life11121297
Reduction in Standard Cancer Screening in 2020 throughout the U.S. Cancers (Basel). 2021 Nov 25;13(23):5918. doi: 10.3390/cancers13235918. PMID: 34885028; PMCID: PMC8656505.
and many others
The first part of the introduction should be deleted because it reports concepts about coronavirus disease that we already know widely.
Please focus the manuscript. on the main or rather CRC and screening
The clinical implications and the limitations of the study must be added
The references section needs to be expanded and more discussion needs to be undertaken, especially about what happened
I suggest the following ref:
DECOR-19 Collaborative Group. DElayed COloRectal cancer care during COVID-19 Pandemic (DECOR-19): Global perspective from an international survey. Surgery. 2021 Apr;169(4):796-807. doi: 10.1016/j.surg.2020.11.008. Epub 2020 Nov 17. PMID: 33353731; PMCID: PMC7670903.
Outcomes from elective colorectal cancer surgery during the SARS-CoV-2 pandemic. Colorectal Dis. 2020 Nov 15:10.1111/codi.15431. doi: 10.1111/codi.15431. Epub ahead of print. PMID: 33191669; PMCID: PMC7753519.
Italian society of colorectal surgery recommendations for good clinical practice in colorectal surgery during the novel coronavirus pandemic. Tech Coloproctol. 2020 Jun;24(6):501-505. doi: 10.1007/s10151-020-02209-6. Epub 2020 Apr 14. PMID: 32291566
and many others
Author Response
Reviewer 2:
This is a review regarding the impact of COVID-19 on Colorectal Cancer Screening.
1. The topic has already been addressed but it is still interesting:
Colorectal Cancer Screening: Impact of COVID-19 Pandemic and Possible Consequences. Life (Basel). 2021 Nov 26;11(12):1297. doi: 10.3390/life11121297
Reduction in Standard Cancer Screening in 2020 throughout the U.S. Cancers (Basel). 2021 Nov 25;13(23):5918. doi: 10.3390/cancers13235918. PMID: 34885028; PMCID: PMC8656505.
and many others
We appreciate the reviewer’s comment. We agree with the reviewer that this topic has been addressed in the past. However, COVID-19 is a rapidly changing and evolving disease, and ours is the most updated literature review about the topic.
2. The first part of the introduction should be deleted because it reports concepts about coronavirus disease that we already know widely.
We appreciate the reviewer’s comment. We shortened the introduction as suggested.
3. Please focus the manuscript. on the main or rather CRC and screening
We appreciate the reviewer’s comment. We have modified the manuscript.
4. The clinical implications and the limitations of the study must be added
We appreciate the reviewer’s comment. We have added a section on clinical implications and limitations to the manuscript.
The references section needs to be expanded and more discussion needs to be undertaken, especially about what happened
I suggest the following ref:
DECOR-19 Collaborative Group. DElayed COloRectal cancer care during COVID-19 Pandemic (DECOR-19): Global perspective from an international survey. Surgery. 2021 Apr;169(4):796-807. doi: 10.1016/j.surg.2020.11.008. Epub 2020 Nov 17. PMID: 33353731; PMCID: PMC7670903.
Outcomes from elective colorectal cancer surgery during the SARS-CoV-2 pandemic. Colorectal Dis. 2020 Nov 15:10.1111/codi.15431. doi: 10.1111/codi.15431. Epub ahead of print. PMID: 33191669; PMCID: PMC7753519.
Italian society of colorectal surgery recommendations for good clinical practice in colorectal surgery during the novel coronavirus pandemic. Tech Coloproctol. 2020 Jun;24(6):501-505. doi: 10.1007/s10151-020-02209-6. Epub 2020 Apr 14. PMID: 32291566
We appreciate the reviewer’s comment. We expanded the discussion of the paper using the above papers referenced as well as the papers suggested below to expand on recent studies regarding CRC screening and the COVID-19 pandemic.
Round 2
Reviewer 2 Report
I'm satisfied with the changes made